# Food Addiction in Eating Disorders: A Cluster Analysis Approach and Treatment Outcome

**DOI:** 10.3390/nu14051084

**Published:** 2022-03-04

**Authors:** Lucero Munguía, Anahí Gaspar-Pérez, Susana Jiménez-Murcia, Roser Granero, Isabel Sánchez, Cristina Vintró-Alcaraz, Carlos Diéguez, Ashley N. Gearhardt, Fernando Fernández-Aranda

**Affiliations:** 1Department of Psychiatry, Universitary Hospital of Bellvitge, 08907 Barcelona, Spain; lmunguia@idibell.cat (L.M.); agaspape10@alumnes.ub.edu (A.G.-P.); sjimenez@bellvitgehospital.cat (S.J.-M.); isasanchez@bellvitgehospital.cat (I.S.); cvintro@bellvitgehospital.cat (C.V.-A.); 2Psychoneurobiology of Eating Disorders and Addictive Behaviors Group, Neurosciences Programme, Bellvitge Biomedical Research Institute (IDIBELL), 08908 Barcelona, Spain; roser.granero@uab.cat; 3Clinical Sciences Department, School of Medicine, Barcelona University, 08907 Barcelona, Spain; 4CIBER Physiopatology, Obesity and Nutrition (CIBERobn), Health Institute Carlos III, 28029 Madrid, Spain; carlos.dieguez@usc.es; 5Department of Psychobiology and Methodology, Autonomous University of Barcelona, 08907 Barcelona, Spain; 6Department of Physiology, CIMUS, Instituto de Investigación Sanitaria, University of Santiago de Compostela, 15782 Santiago de Compostela, Spain; 7Department of Psychology, University of Michigan, Ann Arbor, MI 48109, USA; agearhar@umich.edu

**Keywords:** food addiction, eating disorders, treatment outcome, cluster analysis approach

## Abstract

Background: A first approach of a phenotypic characterization of food addiction (FA) found three clusters (dysfunctional, moderate and functional). Based on this previous classification, the aim of the present study is to explore treatment responses in the sample diagnosed with Eating Disorder(ED) of different FA profiles. Methods: The sample was composed of 157 ED patients with FA positive, 90 with bulimia nervosa (BN), 36 with binge eating disorder (BED), and 31 with other specified feeding or eating disorders (OSFED). Different clinical variables and outcome indicators were evaluated. Results: The clinical profile of the clusters present similar characteristics with the prior study, having the dysfunctional cluster the highest ED symptom levels, the worse psychopathology global state, and dysfunctional personality traits, while the functional one the lowest ED severity level, best psychological state, and more functional personality traits. The dysfunctional cluster was the one with lowest rates of full remission, the moderate one the higher rates of dropouts, and the functional one the highest of full remission. Conclusions: The results concerning treatment outcome were concordant with the severity of the FA clusters, being that the dysfunctional and moderate ones had worst treatment responses than the functional one.

## 1. Introduction

Even though food addiction (FA) has not being included as a formal mental disorder in the Diagnostic and Statistical Manual (DSM-5) [1], it is a concept of ongoing scientific interest and debate. According to the FA model, some foods, especially palatable ones, may be involved in producing both overeating and addictive-like behaviours, thus, phenomenological similarities with addictive disorders could been found [2].

FA has been mentioned as a potential subtype of obesity [3,4,5], and has been associated with Eating Disorders (ED), mainly in binge spectrum disorders as bulimia nervosa (BN) [6,7] and binge eating disorder (BED) [8,9]. It has been associated with higher body mass index (BMI), binge-eating episodes, higher eating psychopathology, more impulsive personality traits, and craving for highly palatable food [10,11,12], as well as poorer response to therapy [13,14].

Additionally, other predictors of developing severe symptomatology of food addiction are presenting dysfunctional personality traits, high emotional dysregulation, and high general psychopathology [15,16], and be women [17].

In a previous study, our group has assessed the heterogeneity within a group of subjects with positive FA (FA+) and have identified differential phenotypes and subgroups among the participants [18] considering general psychopathology, ED severity and personality traits. In the prior study, a sample of ED and obesity patients was included, and three clusters were obtained: (a) dysfunctional cluster (mainly represented by OSFED and BN), (b) moderate cluster (mainly represented by BN and BED patients) and (c) functional cluster (mainly represented by obesity and BED patients).

The obtained results of this study shed some light on the different clinical profiles within patients with ED and obesity who had FA+. However, there is a lack in the literature on how the treatment could be influenced by these severity and cluster groups. To have a deeper understanding of the FA construct, evidence related with treatment outcome could help to fill that gap, by knowing the relationship between treatment outcome, FA, psychopathological dimensions, and other variables.

Thus, based onto the prior study, the aim of the present research is to explore treatment response to Cognitive Behavioural Therapy in the ED sample of the different FA clusters found in the previous study [18]. We hypothesize that the functional cluster, will present better treatment outcomes and lower dropout rates than the moderate and dysfunctional ones. As well, due to belonging to a specific cluster provides information on the patients’ profile in a broad collection of clinical measures, and that the present study only consider ED patients from the original sample, the analysis form the previous research, regarding psychopathological status, the personality traits, ED severity and the diagnostic subtype, will be done as well in this study.

## 2. Materials and Methods

### 2.1. Participants and Procedure

The initial sample was comprised of 234 participants of the original study [18]. The identification of the empirical clusters in this prior research was done through two-step-cluster procedure, using the log-likelihood distance measure (adequate for both quantitative and categorical indicators), and combining a multinomial probability mass function (non-metric data) and a normal density function (metric data). The clustering process was also based on an automatic selection of the number of cluster-classes, based on a large set of indicator variables including the ED severity level, global psychopathological state, personality profile and the diagnostic subtype.

The identification of the empirical clusters in this prior research was done through two-step-cluster procedure, using the log-likelihood distance measure (adequate for both quantitative and categorical indicators), and combining a multinomial probability mass function (non-metric data) and a normal density function (metric data). The clustering process was also based on an automatic selection of the number of cluster-classes, based on a large set of indicator variables including the ED severity level, global psychopathological state, personality profile and the diagnostic subtype, only the ED patients were selected. Therefore, the present study comprises a total of 157 adult women diagnosed with an ED (90 with bulimia nervosa (BN), 36 with binge eating disorder (BED), and 31 with other specified feeding or eating disorder (OSFED)) BN subtype, who presented for treatment to the Eating Disorder Unit within the Department of Psychiatry at Bellvitge University Hospital (HUB) (Barcelona, Spain) from May of 2016 and November 2018. Those patients with Anorexia Nervosa diagnosis were excluded, due to the low prevalence’s of FA in this disorder. All participants included in the study were diagnosed according to the to the DSM-IV-TR criteria [19] through a semi-structured interview with experience clinical psychologist, and diagnoses were reanalysed and recodified post hoc using the DSM-5 criteria [1].

According to the Declaration of Helsinki, the present study was approved by the Clinical Research Ethics Committee (CEIC) of Bellvitge University Hospital, and written informed consent was obtained from all participants. All the assessments were conducted by experienced psychologist and psychiatrists.

### 2.2. Assessment

Alongside the assessment of several clinically relevant variables as age of onset, duration of the disorder, BMI (taking weight and height measures in the first visit to our center by trained staff, using the same device to all patients: Tanita MC 780-S MA portable scale: With segmental multifrequency. Bio Lógica Tecnología Médica SL, Barcelona, Spain), and sociodemographical characteristics such as age, income and marital status, the following Spanish validated instruments were used.

Eating Disorders Inventory 2 (EDI-2) [20], is a self-report questionnaire that assesses different cognitive and behavioural characteristics typical for ED in 11 subscales: Drive for Thinness, Bulimia, Body Dissatisfaction, Ineffectiveness, Perfectionism, Interpersonal Distrust, Interoceptive Awareness and Maturity Fears, Asceticism, Impulse Regulation and Social Insecurity. The measures consists of 91 items, answered on a 6-point Likert scale. The internal consistency of the total scale for our sample was 0.92 (coefficient alpha).

Symptom Checklist-90-Revised (SCL-90-R) [21] validated in Spanish population [22], is a questionnaire used to evaluate a broad range of psychological problems and symptoms of psychopathology considering nine primary symptom dimensions: Somatization, Obsession-Compulsion, Interpersonal Sensitivity, Depression, Anxiety, Hostility, Phobic Anxiety, Paranoid Ideation and Psychoticism; and includes three global indices: global severity index (overall psychological distress), positive symptom distress index (the intensity of symptoms) and a positive symptom total (self-reported symptoms). The global severity index can be used as a summary of the test. The measure consists of 90 items answered on a 5-point Likert scale. The internal consistency of the subscales for our sample range from 0.701 to 0.865, and the global indexes was 0.96 (coefficient alpha).

Temperament and Character Inventory-Revised (TCI-R) [23], validated in Spanish population [24], is a questionnaire that measures four temperament dimensions (Harm Avoidance, Novelty Seeking, Reward Dependence and Persistence) and three character dimensions (Self-Directedness, Cooperativeness and Self-Transcendence) of personality. The measure consists of 240-items and answered on a 5-point Likert scale. The internal consistency of the subscales for our sample range from 0.80 to 0.89 (coefficient alpha).

Yale Food Addiction Scale 2.0 (YFAS2.0) [25], validated in Spanish population [9], is a self-report questionnaire for measuring FA during the previous 12 months. Is based on DSM-5 Criteria and evaluates 11 symptoms and allows the establishment of symptom severity cutoffs: mild (2–3 symptoms), moderate (4–5 symptoms), and severe (6–11 symptoms). The score produce two measurements: (a) a continuous symptom count score that reflects the number of fulfilled diagnostic criteria (ranging from 0 to 11), and (b) a food addiction threshold based on the number of symptoms (at least 2) and self-reported clinically significant impairment or distress. This final measurement allows for the binary classification of food addiction (present versus absent). The measures consist of 35-items answered on a 8-point Likert scale. The internal consistency of the total scale for our sample was 0.93 (coefficient alpha).

### 2.3. Treatment

Patients received cognitive–behavioural therapy (CBT) treatment carried out by experienced psychologists at Bellvitge University Hospital (HUB), which consisted of 16 weekly outpatient group sessions of 90 min each and a follow-up period of 6, 12 and 24 months. A detailed description of the treatment applied could be found in [26], with the following main treatment objectives: cognitive restructuring, problem-solving, emotion management techniques, and normalisation of eating behaviour.

#### Treatment Outcome Assessment

The criterion for dropping out of treatment was not attending three consecutive sessions. Of those patients who completed treatment, the following categorisation, according to their symptomatology, was used: full remission (total absence of ED symptoms for at least 4 consecutive weeks), partial-remission (substantial symptomatic improvement, but with residual symptoms considering DSM-5 criteria. It could be an extinction of behavioural symptoms, as purging, or restriction, but residual cognitive distortions, or intense fear to gain weight, or, vice versa), and non-remission (still meeting full criteria for an ED) [1]. These categories to assess treatment outcome have also been used in prior published studies [27,28]. The assessment of treatment out for this study were performed at the end of the 16 treatment sessions.

### 2.4. Statistical Analysis

Statistical analysis was carried out with Stata17 for Windows (Stata-Corp, College Station, TX, USA) [29]. Firstly, the empirical clusters compared in this study were compared for the measures assessed at baseline, and for the ED diagnostic subtype, to provide information about the specific profile associated to each cluster. Next, the discriminative capacity of the empirical clusters on the treatment response was assessed.

Comparison between the clusters was based on chi-square tests (χ^2^) for categorical variables and with analysis of variance (ANOVA) for quantitative variables. Effect size was calculated with the standardized Cohen-*h* for proportion differences and Cohen-*d* for mean differences (poor effect size was considered for absolute estimates lower than 0.50, mild-moderate effect size for absolute estimates higher than 0.50 and high-large for absolute estimates higher than 0.80) [30]. Control in Type-I error due to the multiple null-hypothesis statistical tests was done with Finner’s method [31].

## 3. Results

Most participants in the study were single (*n* = 97, 61.8%) and had achieved secondary education levels (*n* = 74, 47.1%). Mean age was 33.2 years (SD = 11.9), mean age of onset of the eating problems was 21.1 years (SD = 9.4) and mean duration of the eating problems was 12.3 years (SD = 9.1). Table 1 includes the comparison between the groups for the descriptive variables. Considering the groups defined by the ED-subtype, BED included patients with the highest age and the oldest age of onset. Comparison between the clusters identified differences for age and onset for the dysfunctional cluster (C1), which had lower means compared to the moderate (C2) and the functional (C3) clusters.

Figure 1 displays the 100% stacked bar chart with the percentage of patients with a specific ED subtype within each cluster. Differences between the groups were found: The dysfunctional cluster (C1) included a high and similar distribution for BN and OSFED patients; the moderate (C2) cluster included mostly BN patients, following by a high percentage as well of BED; the functional (C3) cluster included a high proportion of BN patients, and similar percentage of BED and OSFED.

The upper part of Table 2 shows the comparison between the clusters at baseline, and the lower part of the table shows the comparison for the CBT treatment outcomes. FA levels was higher in the moderate cluster (C2), followed by the dysfunctional one (C1), while the C3 (functional) presented the lower levels of FA. According to clinical characteristics, the dysfunctional cluster (C1) was characterized by the lowest mean for the BMI, the highest ED symptom levels (except for the EDI-2 bulimia scale), the worst psychopathology global state, and the highest levels in the personality domains of harm avoidance and self-transcendence. This cluster was also the one with the lowest percentage of participant with full remission (see also Figure 2). The functional cluster (C3) was the cluster with the lowest ED severity level, best psychological state, the lowest score in harm avoidance, and the highest scores in the personality traits of reward-dependence, persistence, self-directedness and cooperativeness. As well, this cluster also had the highest percentage of patients with full remission (Figure 2). C2, the moderate one, present intermediate levels of these clinical characteristics; however, it had the highest levels of dropouts.

## 4. Discussion

The aim of the present study was to explore treatment responses in the different FA profiles identified by [18], considering only the ED sample. Clinical characteristics of these ED-focused clusters are similar to those previously found and were relevant for treatment outcome as well. As we hypothesized, the functional cluster (C3), do present better treatment response and lower dropout rates than the moderate (C2) and dysfunctional (C1) clusters. Several aspects of these results must be highlighted.

First, as in the prior study [18], FA levels were higher in the moderate cluster, followed by the dysfunctional one, and lower in the functional. The composition of each cluster regarding the diagnosis of the patients was maintained for the dysfunctional and moderate clusters, however, the composition of the functional one changed. In the prior study, this subgroup was highly represented by patients with obesity but no ED, while in this study non-ED participants were excluded. However, the clinical characteristics of the present and previous clusters were similar. This is, dysfunctional cluster (C1) had higher presence of BN and OSFED patients, higher severity of the disorder and worst psychopathological state, as well as low self-directedness and high harm avoidance. The functional cluster (C3) had more equilibrated proportion of diagnosis subtypes, with BN being more prevalent, and higher self-directedness and persistence, with lower levels of harm-avoidance. Finally, the moderate cluster (C2) had a heightened presence of BN (72.5%) followed by BED (24.6%), therefore, this cluster was particularly represented by binge ED subtypes; as well, this cluster had the highest levels of FA as in the first study.

Thus, what differentiates the dysfunctional cluster (C1) from the other clusters is the severity of it clinical characteristics (except FA), while the moderate (C2) group differs from the functional (C3) and dysfunctional cluster (C1) by a higher severity of FA, and the functional cluster (C3) differ from the dysfunctional (C1) and moderate (C2) one by the low severity of its clinical profile.

Treatment outcome was explored as well, not only to relate it with the presence of FA, being that other studies have already approach the subject [14,32], but to add to a better characterization of FA construct in well-defined phenotypes that consider FA presence and other clinical variables.

Low levels of full remission and higher rates of dropouts in the dysfunctional cluster (C1) were found. This subgroup was highly represented for OSFED patients, which have been reported to present low harm avoidance and self-directedness, as well as higher severity of ED symptomatology, aspects identified as predictors of high drop-outs and low full remission rates [28]. Additionally, similar personality traits that imply difficulties in following goals and higher levels of anxiety levels have been found in BN patients (also present in this cluster) [33]. This has also been associated with low levels of full remission after cognitive behavioural treatment (CBT) [26]. Therefore, patients within this cluster may benefit from treatments that target the reduction of the ED symptomatology and general distress, as well as favour the improvement in the establishment and following of objectives. It is also important to mention that younger patients with an earlier onset of the disorder were particularly present in this cluster; therefore, these aspects could be added as indicators of a more dysfunctional profile. Of note, early onset of the disorder has already been mentioned as a predictor of a longer maintenance of the ED [34].

In the moderate cluster (C2), the highest dropout rates were found, as well medium rates of full remission in comparison with the dysfunctional (C1) and functional clusters (C3). This cluster was characterized by the presence of binge spectrum ED patients and by the higher levels of FA, both aspects that could be involved in the response to treatment of the participants in this cluster. It is possible that the higher presence of FA symptomology in binge spectrum ED (relative to non-binge ED) may reflect the more frequent binge eating episodes and food craving associated with FA [35,36]. In the same line, it could be hypostatized that the high levels of FA could be related to the higher drop out percentage found here. This is consistent with prior studies that have found that FA predicts worse intervention response in BN patients [32], and that FA can act as a mediator between severity of ED in BN and BED patients and treatment outcome [14]. However, being that this cluster did not present levels of psychopathology and severity of the ED as high as the dysfunctional cluster (C1), this moderate cluster (C2) was more likely to reach full remission than the patients in C1. Therefore, it may be important to screen for severe FA (particularly for patients with diagnoses more represented in this cluster), to implement approaches that could reduce FA symptomatology. In this regard, several authors have suggested additional therapy aims of craving managements and increasing inhibitory control [37], and psychoeducation about the dietary patterns implicated in addictive eating [35,38]. As well, BED patients and FA patients present high levels of impulsivity, therapies aimed to reduce food related impulsivity could be implemented as well [39].

Finally, the functional cluster (C3) presents the higher levels of full remission and the lowest of drop out, as well as had the lowest levels of severity of FA. Further, this cluster is clinically speaking the most functional, presenting low ED severity and general psychopathology. It also had the highest levels of self-directedness and persistence, which may be associated with good compliance with the treatment. Patients within this subgroup may respond best to traditional CBT treatment and not need additive therapies to help them succeed. This groups overall FA severity level is low and other studies have found that FA symptoms can remit after traditional CBT [32]. Thus, specific targets for addictive mechanisms may not be needed for this cluster.

## 5. Strength and Limits

The present study has several strengths. Not only the relation of FA with ED treatment outcome was explored, a characterization of different phenotypes in ED patients that presents FA was confirmed. In this sense, it is important to consider that the clustering process was performed in the original study for a large set of indicators measured in the baseline (previously to the intervention), which included the ED severity level along with other variables (the comorbid psychopathological status, the personality traits and the diagnostic subtype). In this sense, belonging to a specific cluster provides wide information on the patients’ profile in a broad collection of clinical measures. Since this work provides results for the comparison between these empirical profiles, this study can be conceptualized within a person-centred approach, characterized by the analysis of individuals who share multiple particular attributes (contrarily to the classical variable-centred approach, focused on assessing the specific contribution of isolated variables). Additionally, these clusters have specific clinical characteristics, which may inform precision therapies.

Even so, some limitations may be taken into account. The study was only performed in women patients, and the size of the sample was considerably reduced due to the exclusion of patients with obesity only (as the aim of this paper was on ED treatment response). These aspects affect the generalization of the results, and future studies should consider them.

## 6. Conclusions

The phenotypic characterization of FA proposed by the original study [18] (considering ED and obesity patients), and the present one (only considering ED patients), present similar clinical characteristics that again defined three clusters from a dysfunctional to functional one. The dysfunctional cluster had the lower rates in full remission, while the functional cluster had the higher proportions of full remission and the lowest of dropouts. Even though the moderate cluster presented the highest rate of dropouts, a higher percentage of participants in this cluster could reach full remission of their symptoms. Even though all participants presented FA symptoms, the differential characteristics of each cluster may be important to defining proper treatment approaches for ED patients with FA. For example, the dysfunctional cluster may benefit from treatments that target aspects of high severity ED symptoms and psychological distress; the moderate cluster may be specifically benefited by a focused treatment for the reduction of FA symptoms; finally, the functional cluster could continue with traditional approaches.

## Figures and Tables

**Figure 1 nutrients-14-01084-f001:**
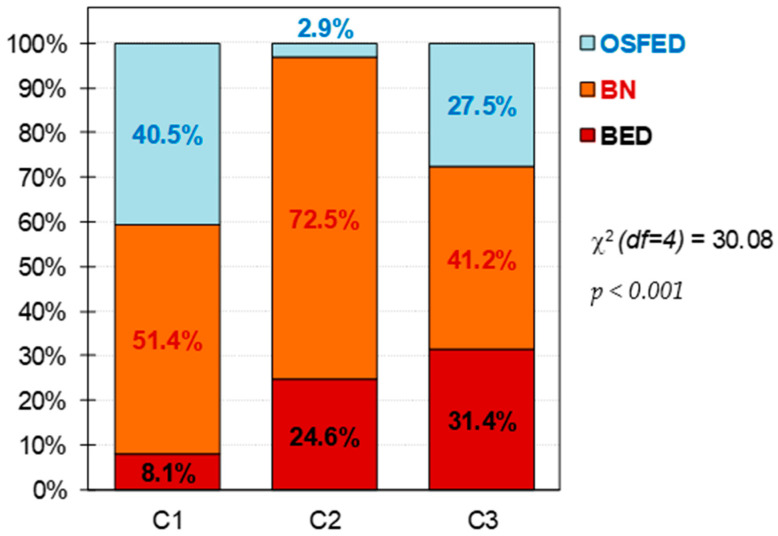
Composition of the clusters. Note. C1: cluster 1, dysfunctional cluster. C2: cluster 2, moderate cluster. C3: cluster 3, functional cluster. BED: binge eating disorder. BN: bulimia nervosa. OSFED: other specified feeding eating disorder. df = degrees of freedom. Sample size: *n* = 157.

**Figure 2 nutrients-14-01084-f002:**
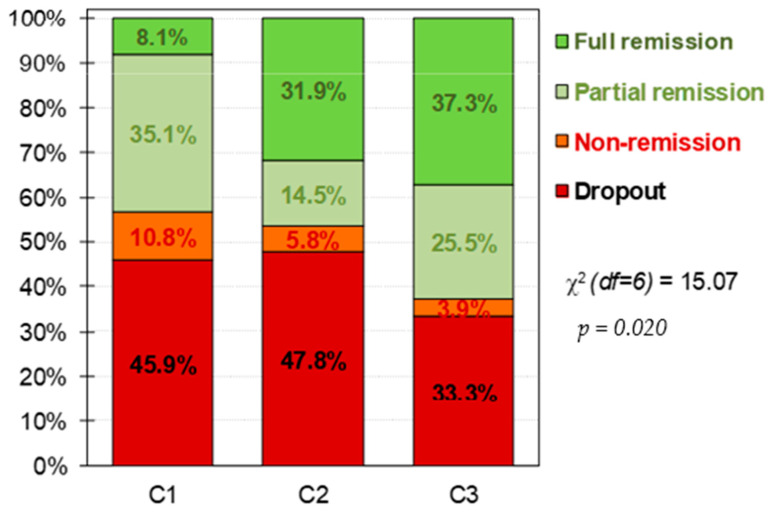
Distribution of the CBT outcomes within the clusters. Note. C1: cluster 1, dysfunctional cluster. C2: cluster 2, moderate cluster. C3: cluster 3, functional cluster. df = degrees of freedom. Sample size: *n* = 157.

**Table 1 nutrients-14-01084-t001:** Descriptive of the sample.

	BED (*n* = 36)		BN (*n* = 90)		OSFED (*n* = 31)			Cluster-1 (*n* = 37)		Cluster-2 (*n* = 69)		Cluster-3 (*n* = 51)		
	*n*	*%*	*n*	*%*	*n*	*%*	*p*	*n*	*%*	*n*	*%*	*n*	*%*	*p*
Civil status														
Single	14	38.9%	57	63.3%	26	83.9%	**0.005 ***	29	78.4%	41	59.4%	27	52.9%	0.144
Married	16	44.4%	22	24.4%	3	9.7%		6	16.2%	20	29.0%	15	29.4%	
Divorced	6	16.7%	11	12.2%	2	6.5%		2	5.4%	8	11.6%	9	17.6%	
Education														
Primary	13	36.1%	31	34.4%	11	35.5%	0.688	13	35.1%	25	36.2%	17	33.3%	0.629
Secondary	14	38.9%	44	48.9%	16	51.6%		20	54.1%	32	46.4%	22	43.1%	
University	9	25.0%	15	16.7%	4	12.9%		4	10.8%	12	17.4%	12	23.5%	
	Mean	SD	Mean	SD	Mean	SD	*p*	Mean	SD	Mean	SD	Mean	SD	*p*
Age (years-old)	42.28	12.30	30.92	10.39	29.13	10.41	**0.001 ***	28.14	8.86	34.74	12.89	34.71	11.62	**0.012 ***
Onset (years-old)	27.72	11.45	18.98	7.89	19.29	7.44	**0.001 ***	17.84	6.41	22.57	11.38	21.32	7.81	**0.046 ***
Duration (years)	14.46	9.16	12.13	9.00	10.03	8.84	0.134	10.32	7.49	12.80	9.84	12.91	8.99	0.334

Note. BED: binge eating disorder. BN: bulimia nervosa. OSFED: other specified feeding and eating disorder. SD: standard deviation. Custer 1: dysfunctional cluster; Cluster 2: Moderate cluster; Cluster 3: Functional cluster. * Bold: significant comparison (0.05).

**Table 2 nutrients-14-01084-t002:** Comparison of clusters at baseline and CBT outcomes.

		Cluster-1 (*n* = 37)		Cluster-2 (*n* = 69)		Cluster-3 (*n* = 51)		Cluster-1 vs. Cluster-2		Cluster-1 vs. Cluster-3		Cluster-2 vs. Cluster-3	
	*α*	Mean	SD	Mean	SD	Mean	SD	*p*	*|d|*	*p*	*|d|*	*p*	*|d|*
BMI-FA													
BMI (kg/m^2^)		25.96	7.44	29.42	8.54	30.77	10.15	0.057	0.43	**0.013 ***	**0.54 ^†^**	0.411	0.14
YFAS total score	0.939	8.46	2.38	9.48	1.99	7.53	2.72	**0.034 ***	0.46	0.068	0.36	**0.001 ***	**0.82 ^†^**
EDI -2 Drive-thinness	0.767	18.03	2.71	15.94	4.77	14.14	4.94	**0.022 ***	**0.54 ^†^**	**0.001 ***	**0.98 ^†^**	**0.029 ***	0.37
EDI-2 Body-dissatisfac.	0.850	21.30	5.73	20.59	6.52	16.96	7.17	0.600	0.11	**0.003 ***	**0.67 ^†^**	**0.003 ***	**0.53 ^†^**
EDI-2 Int-awareness	0.798	18.22	5.67	15.46	5.36	8.00	5.71	**0.016 ***	**0.50 ^†^**	**0.001 ***	**1.80 ^†^**	**0.001 ***	**1.35 ^†^**
EDI-2 Bulimia	0.726	8.54	5.78	11.52	3.91	7.33	4.89	**0.002 ***	**0.60 ^†^**	0.239	0.23	**0.001 ***	**0.95 ^†^**
EDI-2 Interper-distrust	0.813	9.08	5.24	6.97	4.65	3.49	3.60	**0.022 ***	0.43	**0.001 ***	**1.24 ^†^**	**0.001 ***	**0.84 ^†^**
EDI-2 Ineffectiveness.	0.848	17.38	6.55	14.88	5.70	6.88	4.68	**0.031 ***	0.41	**0.001 ***	**1.84 ^†^**	**0.001 ***	**1.53 ^†^**
EDI-2 Maturity-fears	0.752	12.27	5.03	9.17	5.32	6.51	5.17	**0.004 ***	**0.60 ^†^**	**0.001 ***	**1.13 ^†^**	**0.006 ***	**0.51 ^†^**
EDI-2 Perfectionism	0.740	6.95	5.12	6.14	4.24	4.65	3.97	0.371	0.17	**0.016 ***	**0.50 ^†^**	0.066	**0.36**
EDI-2 Impulse-regulat.	0.730	13.22	5.28	7.57	4.37	3.18	3.18	**0.001 ***	**1.17 ^†^**	**0.001 ***	**2.30 ^†^**	**0.001 ***	**1.15 ^†^**
EDI-2 Ascetic	0.702	10.35	2.99	8.77	2.92	5.61	3.11	**0.010 ***	**0.54 ^†^**	**0.001 ***	**1.56 ^†^**	**0.001 ***	**1.05 ^†^**
EDI-2 Social Insecurity	0.752	12.76	4.78	9.41	4.17	4.49	2.82	**0.001 ***	**0.75 ^†^**	**0.001 ***	**2.11 ^†^**	**0.001 ***	**1.38 ^†^**
EDI-2 Total score	0.923	148.1	27.28	126.4	20.73	81.24	22.63	**0.001 ***	**0.89 ^†^**	**0.001 ***	**2.67 ^†^**	**0.001 ***	**2.08 ^†^**
SCL-90R GSI	0.966	2.67	0.33	2.07	0.35	1.28	0.36	**0.001 ***	**1.80 ^†^**	**0.001 ***	**4.03 ^†^**	**0.001 ***	**2.22 ^†^**
SCL-90R PST	0.966	81.81	6.10	72.46	7.78	55.98	11.90	**0.001 ***	**1.34 ^†^**	**0.001 ***	**2.73 ^†^**	**0.001 ***	**1.64 ^†^**
SCL-90R PSDI	0.966	2.94	0.33	2.58	0.36	2.04	0.34	**0.001 ***	**1.05 ^†^**	**0.001 ***	**2.70 ^†^**	**0.001 ***	**1.54 ^†^**
TCI-R Novelty-seeking	0.806	103.5	16.07	98.4	17.27	102.7	15.88	0.133	0.31	0.811	0.05	0.168	0.26
TCI-R Harm-avoidance	0.887	133.7	14.52	126.4	17.00	109.0	16.24	**0.028 ***	0.46	**0.001 ***	**1.60 ^†^**	**0.001 ***	**1.05 ^†^**
TCI-R Reward.depend.	0.831	97.5	17.30	98.3	14.06	104.8	15.59	0.797	0.05	**0.029 ***	0.44	**0.023 ***	0.44
TCI-R Persistence	0.896	102.8	22.46	100.8	20.27	108.4	19.78	0.633	0.09	0.213	0.26	**0.048 ***	0.38
TCI-R Self-directed.	0.840	96.9	14.94	102.9	13.17	125.3	16.89	0.053	0.42	**0.001 ***	**1.78 ^†^**	**0.001 ***	**1.48 ^†^**
TCI-R Cooperativeness	0.861	127.8	20.24	133.7	17.15	139.3	11.88	0.082	0.31	**0.002 ***	**0.69 ^†^**	0.067	0.38
TCI-R Self-transcend.	0.862	77.1	12.09	62.1	14.38	63.2	16.37	**0.001 ***	**1.13 ^†^**	**0.001 ***	**0.97 ^†^**	0.672	0.07
CBT outcomes		*n*	*%*	*n*	*%*	*n*	*%*	*p*	*|h|*	*p*	*|h|*	*p*	*|h|*
Dropout		17	45.9%	33	47.8%	17	33.3%	**0.010 ***	0.04	**0.016 ***	0.26	0.286	0.30
Non-remission		4	10.8%	4	5.8%	2	3.9%		0.18		0.27		0.09
Partial remission		13	35.1%	10	14.5%	13	25.5%		**0.51 ^†^**		0.21		0.28
Full remission		3	8.1%	22	31.9%	19	37.3%		**0.62 ^†^**		**0.74 ^†^**		0.11

Note. Cronbach’s-alpha in the study. Custer 1: dysfunctional cluster; Cluster 2: Moderate cluster; Cluster 3: Functional cluster. SD: standard deviation; BMI: Body Mass Index; FA: food addiction; EDI: Eating Disorders Inventory; SCL: Symptom Checklist; GSI: Global Gravity Index; TCI: Temperament and Character Inventory; CBT: cognitive–behavioural therapy. * Bold: significant comparison (0.05). ^†^ Bold: effect size into the ranges mild-moderate to the high-large.

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
