# Peer review of "Food Addiction in Eating Disorders: A Cluster Analysis Approach and Treatment Outcome"

_nutrients, 2022, doi:10.3390/nu14051084_

Round 1

Reviewer 1 Report

nutrients-1589854 Food Addiction in Eating Disorders. A cluster analysis approach and treatment outcome

I would like to thank the authors for the opportunity to review their manuscript. This project aimed to investigate the impact of food addiction on outcome from group cognitive-behavioral treatment for eating disorders (binge eating disorder, bulimia nervosa, OSFED). The authors grouped participants into one of 3 pre-determined food addiction clusters and found that drop out and symptom remission at the end of treatment differed across the 3 food addiction profiles. This is an interesting approach to elucidating how food addiction may impact treatment outcome among individuals with binge-eating spectrum eating disorders. However, additional details are needed to improve the interpretability of the study findings.

Overall, more information is needed regarding how the 3 food addiction clusters were derived. It is understood that the clustering analyses were published in a prior manuscript, however, this second manuscript should provide the reader with enough information to understand the methodology used without having to read the prior paper. Additionally, it is unclear whether the clusters provide added value for predicting treatment outcome. The analytic approach utilized in the present manuscript only offers a cursory indication that patterns of outcome differ among the 3 food addiction clusters. It seems relevant to test whether these clusters offer predictive value above and beyond the severity of food addiction, or other factors such as initial eating disorder symptom severity. Indeed, the functional cluster presented with lower severity at baseline as well as greater remission at the end of treatment. The cluster that presented with the highest food addiction severity had the highest rates of drop-out. Therefore, it is unclear whether the food addiction clusters or initial food addiction and eating disorder symptoms are driving the differential outcomes.

Additional recommendations are provided below.

Abstract

  • The first sentence of the abstract was a bit hard to understand.

Introduction

  • Inconsistencies in grammar and tense used throughout the introduction made it difficult to discern what research has already been conducted and what the authors think are remaining questions for study. For example, line 55- should be “identified” instead of identify.
  • Line 50- The way in which this sentence is written seems to suggest that females have dysfunctional personalities and are emotionally volatile. I do not believe the authors intended to voice negative stereotypes about females, but it is recommended that they rephrase this sentence.
  • Lines 54-60- As mentioned above, more information is needed regarding how these clusters were determined.
  • More information supporting why the predetermined clusters are likely to have added predictive validity above and beyond the presence and severity of food addiction symptoms could bolster the rationale for the present study.

Methods/Materials

  • Could the authors clarify whether other inclusion/exclusion criteria were used in the present study? For example, it appears that the authors did not include participants meeting criteria for Anorexia Nervosa, was that an exclusion criterion?
  • Could the authors clarify whether all OSFED diagnoses or only a subset were included in the present study? For example, were individuals with night feeding or purging disorder included?
  • Additional clarification on the assessment of treatment outcome is needed.
    • It is unclear whether treatment outcome was assessed at the end of treatment, or at one of the various follow-ups timepoints
    • If a participant did not attend 3 sessions but returned afterward, were they not provided additional therapy? If they were provided additional therapy, were they considered to have dropped out, or were they included in the other treatment outcome groups?
    • Line 141, could the authors clarify how they operationalized “substantial symptomatic improvement”
    • If it occurred, could the authors clarify how they handled diagnostic cross-over when categorizing remission? For example, if a patient initially presented with BN, then ceased purging but met full criteria for BED at the end of treatment, were they considered to be in full, partial, or no remission?

Line 147/148- Could the authors specify whether they are referring to comparing across diagnostic groups or food addiction clusters.

Results

  • It can be helpful to understand whether patients also presented with comorbid diagnoses. Would the authors be willing to include these data?
  • It would be helpful if the authors could provide a breakdown of the various OSFED diagnoses present in their sample.
  • Did the authors collect data on the reason for drop out? Having additional context on the reason for early treatment termination might help the reader better understand the link between food addiction severity and drop out?
  • Given that the only stated aim of the present study was to determine treatment outcome across the 3 food addiction clusters, it was surprising to see additional analyses conducted comparing both food addiction clusters and ED diagnostic groups on personality traits and other clinical characteristics. The rationale for conducting these analyses, as well as a-priori hypotheses, should be added to the introduction.
  • There is some concern regarding the replicability of the clusters in the present sample. For example, food addiction was highest in the “moderate” cluster which was presented as the middle severity cluster. If the goal is to improve precision therapies, it would seem to be important to derive food addiction clusters from the specific clinical population of interest, rather than extrapolating groups from a different population. The rationale for utilizing previously derived clusters rather than deriving clusters from a sample of adults with eating disorders could be strengthened.
  • The inclusion of analyses comparing ED diagnostic groups, irrespective of food addiction cluster, appeared to detract from the comparisons across clusters and made it unclear what additional value the clusters provide.
  • Drop out seems qualitatively different from completing treatment. It is also more common that drop out and remission are examined in separate analyses. Therefore, the authors could bolster their rationale for including treatment drop-out in the same analyses as treatment remission.
  • According to Figure 2, it appears that a large percentage (close to 50%) of the sample dropped out prior to the end of treatment. Could the authors provide an overall sample breakdown of how many patients were included in each outcome group? This could help the reader better understand how treatment outcome in the present study might compare to prior studies.

Discussion

  • It appears that the 3 food addiction clusters did not replicate perfectly from the prior study. For example, food addiction was highest in the “moderate” cluster and did not significantly differ between the highest and lowest severity clusters. This calls to question the replicability and generalizability of these previously derived clusters to an eating disorder population. It is possible that an improved understanding of how the clusters were originally derived would mitigate this concern. Currently, discussion as to why the food addiction clusters may not have replicated seems warranted.
  • Given the pattern of findings, it is unclear whether the clusters are driving treatment outcome. Rather, it appears that food addiction severity maps onto treatment drop out and patients with lower symptom severity at the start of treatment achieve full remission. These findings seem to diminish the utility of the food addiction clusters. More intensive analyses testing the added clinical and predictive value of the clusters, above and beyond these factors, could increase confidence in the author’s conclusions.
  • It appears that one major limitation of the present study is the lack of a control group of individuals with eating disorders who do not screen positive for food addiction. The addition of this group could have provided more confidence that food addiction clustering, rather than eating disorder diagnosis or symptom severity, impacts treatment outcome.

Author Response

February 25th, 2022

Dear Editor

Thank you for your and the reviewers’ comments. As suggested, we have attached a revised version of the manuscript entitled “Food Addiction in Eating Disorders. A cluster analysis approach and treatment outcome” (Manuscript ID: nutrients-1589854).

We have made changes to the manuscript according to the reviewers’ comments, using the tool: control changes, as well, relevant changes have been written as answer´s to the comments reviewers. The manuscript has been prepared according to the journal's instructions. Our answers to the reviewers can be seen below.

Reviewers' comments:

Reviewer #1:

I would like to thank the authors for the opportunity to review their manuscript. This project aimed to investigate the impact of food addiction on outcome from group cognitive-behavioral treatment for eating disorders (binge eating disorder, bulimia nervosa, OSFED). The authors grouped participants into one of 3 pre-determined food addiction clusters and found that drop out and symptom remission at the end of treatment differed across the 3 food addiction profiles. This is an interesting approach to elucidating how food addiction may impact treatment outcome among individuals with binge-eating spectrum eating disorders. However, additional details are needed to improve the interpretability of the study findings.

Response. On behalf of my co-authors, we greatly appreciate the Reviewer’s positive remarks and suggestions on our manuscript. We have considered all the comments and incorporated them into the revised manuscript. Changes to the original document are highlighted as red-color-font, and an itemized point-by-point response to the Reviewer is presented below.

Overall, more information is needed regarding how the 3 food addiction clusters were derived. It is understood that the clustering analyses were published in a prior manuscript, however, this second manuscript should provide the reader with enough information to understand the methodology used without having to read the prior paper.

Response. Thank you for pointing this out. We have now included the clustering procedure used in the original study:

“The identification of the empirical clusters in this prior research was done through two-step-cluster procedure, using the log-likelihood distance measure (adequate for both quantitative and categorical indicators), and combining a multinomial probability mass function (non-metric data) and a normal density function (metric data). The clustering process was also based on an automatic selection of the number of clus-ter-classes, based on a large set of indicator variables including the ED severity level, global psychopathological state, personality profile and the diagnostic subtype”

Additionally, it is unclear whether the clusters provide added value for predicting treatment outcome. The analytic approach utilized in the present manuscript only offers a cursory indication that patterns of outcome differ among the 3 food addiction clusters. It seems relevant to test whether these clusters offer predictive value above and beyond the severity of food addiction, or other factors such as initial eating disorder symptom severity. Indeed, the functional cluster presented with lower severity at baseline as well as greater remission at the end of treatment. The cluster that presented with the highest food addiction severity had the highest rates of drop-out. Therefore, it is unclear whether the food addiction clusters or initial food addiction and eating disorder symptoms are driving the differential outcomes.

Response. Thank you for highlighting this. The objective of this study was to assess the discriminative capacity of the empirical clusters on the treatment outcome. It is important to highlight that the clustering process was originally performed for a large set of indicators measures in the baseline (previously to the intervention), which included precisely the ED severity level along with other variables (the comorbid psychopathological status, the personality traits and the diagnostic subtype). In this sense, belonging to a specific cluster provides wide information about the patients’ "profile" in a broad group of clinical indicators. This is the reason for performing crude comparisons between the three empirical clusters, without adjusting for potential covariates which were previously used during the clustering. That is, to "isolate" specific variables in the comparison process would not be the best option, since we are not interested in a variable-centered approach (focused on the predictive capacity of specific variables), but in a person-centered approach (focused on the comparison of groups of individuals who share particular attributes). We have clarified this issue in the discussion section, since it is precisely one strength of this work:

“The present study has several strengths. Not only the relation of FA with ED treatment outcome was explored, a characterization of different phenotypes in ED patients that presents FA was confirmed. In this sense, it is important to consider that the clustering process was performed in the original study for a large set of indicators measured in the baseline (previously to the intervention), which included the ED severity level along with other variables (the comorbid psychopathological status, the personality traits and the diagnostic subtype). In this sense, belonging to a specific cluster provides wide information on the patients’ profile in a broad collection of clinical measures. Since this work provides results for the comparison between these empirical profiles, this study can be conceptualized within a person-centered approach, characterized by the analysis of individuals who share multiple particular attributes (contrarily to the classical variable-centered approach, focused on assessing the specific contribution of isolated variables)”.

Additional recommendations are provided below.

Abstract

Comment 1:

The first sentence of the abstract was a bit hard to understand.

Response: Thank you so much for make us notice it, we have changed the sentence in the abstract as follows:

A first approach of a phenotypic characterization of Food Addiction (FA) found 3 clusters (dysfunctional, moderate and functional). Based on this previous classification, the aim of the present study is to explore treatment responses in the sample diagnosed with Eating Disorder of different FA profiles.

Introduction

Comment 2:

Inconsistencies in grammar and tense used throughout the introduction made it difficult to discern what research has already been conducted and what the authors think are remaining questions for study. For example, line 55- should be “identified” instead of identify.

Response: Thank for your comment, this has been reviewed and modified along the introduction.

Comment 3:

Line 50- The way in which this sentence is written seems to suggest that females have dysfunctional personalities and are emotionally volatile. I do not believe the authors intended to voice negative stereotypes about females, but it is recommended that they rephrase this sentence.

Response: Thank you so much for make us notice this, we have changed the paragraph as follows:

Additionally, other predictors of develop severe symptomatology of food addiction are to present dysfunctional personality traits, high emotional dysregulation, and high general psychopathology (Brunault et al., 2018; Burrows, Kay-Lambkin, Pursey, Skinner, & Dayas, 2018), and be women (Burrows, Skinner, McKenna, & Rollo, 2017).

Comment 4:

Lines 54-60- As mentioned above, more information is needed regarding how these clusters were determined.

Response. Thank you for this constructive remark. We have clarified this issue in the methods section (see our first response for the overall comments).

Comment 5:

More information supporting why the predetermined clusters are likely to have added predictive validity above and beyond the presence and severity of food addiction symptoms could bolster the rationale for the present study.

Response. We appreciate this helpful suggestion. We have also clarified this issue in the discussion section (see also our second response for the overall comments).

Methods/Materials

Comment 6:

Could the authors clarify whether other inclusion/exclusion criteria were used in the present study? For example, it appears that the authors did not include participants meeting criteria for Anorexia Nervosa, was that an exclusion criterion?

Response: Thank you so much for given us the opportunity to clarify this, we have added the following sentence to the participants and procedure section.

Those patients with Anorexia Nervosa diagnosis were excluded, due to the low prevalence’s of FA in this disorder.

Comment 7:

Could the authors clarify whether all OSFED diagnoses or only a subset were included in the present study? For example, were individuals with night feeding or purging disorder included?

Response: We have clarified this information in the participants and procedure section. Only those patients with BN OSFED subtype have been included, due to the same reason of the previous comment, there is a low low prevalence’s of FA in restrictive eating disorders.

Comment 8:

Additional clarification on the assessment of treatment outcome is needed.

Response: The information regarding this topic in included in the treatment section, and we have added a subheading pointed out the treatment outcome assessment.

Comment 9:

It is unclear whether treatment outcome was assessed at the end of treatment, or at one of the various follow-ups timepoints

Response: Thank you for give us the opportunity to add this important data, we have clarified this in the treatment outcome assessment section as follows:

The assessment of treatment out for this study were performed at the end of the 16 treatment sessions.

Comment 10:

If a participant did not attend 3 sessions but returned afterward, were they not provided additional therapy? If they were provided additional therapy, were they considered to have dropped out, or were they included in the other treatment outcome groups?

Response: If this happened, the patients do not return to the treatment they dropped out, or receive any other at that time, they start again the process, some of them have to be at waiting list, and then they can receive treatment again, but from the beginning.

Comment 11:

Line 141, could the authors clarify how they operationalized “substantial symptomatic improvement”

Response: Thank you for the comment, we have added how could it be considered as follows:

Substantial symptomatic improvement but with residual symptoms considering DSM-5 criteria. It could be, an extinction of behavioral symptoms, as purging, or restriction, but, residual cognitive distortions, or intense fear to gain weight, or, vice versa

Comment 12:

If it occurred, could the authors clarify how they handled diagnostic cross-over when categorizing remission? For example, if a patient initially presented with BN, then ceased purging but met full criteria for BED at the end of treatment, were they considered to be in full, partial, or no remission?

Response: Thank you for given us the opportunity to clarified this. Even if it is possible, and recurrent that crossover ED diagnosis subtype could exist, this do not happen after the remission of one symptom, due to each ED subtype is characterized for other cognitive components, evolution of the disorder, risk factors, or other associated variables. In this sense, a BN that stop purging, do not perse will become a BED, that is why it is continue treated as a BN trough the end of the treatment or the follow ups.

Comment 13:

Line 147/148- Could the authors specify whether they are referring to comparing across diagnostic groups or food addiction clusters.

Response: In order to clarified this, the word groups, have been changed for clusters.

Results

Comment 14:

It can be helpful to understand whether patients also presented with comorbid diagnoses. Would the authors be willing to include these data?

Response: Thank you very much for this suggestion, we highly appreciated, but unfortunately, it is not possible to include this information, however, for future research studies we will consider it.

Comment 15:

It would be helpful if the authors could provide a breakdown of the various OSFED diagnoses present in their sample.

Response: Please see the answer to comment 7.

Comment 16:

Did the authors collect data on the reason for drop out? Having additional context on the reason for early treatment termination might help the reader better understand the link between food addiction severity and drop out?

Response: Thank you for your comment, the highest reason of the dropout, trough, clinical experience, are: from 0 to 3 weeks, lack of motivation, bad adaptation to the treatment, failure to achieve therapeutic alliance; and, from 3 to further weeks, dropouts tent to be related with other external motives, as change of work, to move to another city or country, among others. For sure, this information will be consider for future studies.

Comment 17:

Given that the only stated aim of the present study was to determine treatment outcome across the 3 food addiction clusters, it was surprising to see additional analyses conducted comparing both food addiction clusters and ED diagnostic groups on personality traits and other clinical characteristics. The rationale for conducting these analyses, as well as a-priori hypotheses, should be added to the introduction.

Response: Thank you for make us notice this, we have added the following paragraph at the end of the introduction, following your suggestion.

As well, due to belonging to a specific cluster provides information on the patients’ profile in a broad collection of clinical measures, and that the present study only consider ED patients from the original sample, the analysis form the previous research, regarding psychopathological status, the personality traits, ED severity and the diagnostic subtype, will be done as well in this study.

Comment 18:

There is some concern regarding the replicability of the clusters in the present sample. For example, food addiction was highest in the “moderate” cluster which was presented as the middle severity cluster. If the goal is to improve precision therapies, it would seem to be important to derive food addiction clusters from the specific clinical population of interest, rather than extrapolating groups from a different population. The rationale for utilizing previously derived clusters rather than deriving clusters from a sample of adults with eating disorders could be strengthened.

Response: Thank you very much for your suggestions, we have already strength the rationality in several points of the article considering your suggestions, for example, in the introduction, discussion and limits and strengths. As well, we would like to mention again, that the aim of the present study was, precisely, in an already identify population with FA, with a phenotype characterization done, to complete this characterization with the knowledgement of the treatment outcomes in this population. About the replicability, in the first study, the moderate cluster was the one with highest FA as well as in this one. All the participants have FA, but the clusters severity was defined by FA and other clinical characteristics as well. The moderate cluster was labelled like that considering the whole analysis of the variables, not only FA. What it was demonstrated was that ED patients with FA positive could grouped in different clusters and, through their characteristics, to find the best treatment proposal, in order to complete this important aim, the present research was done, and the treatment outcome results have give us information of the best treatment options, how it is exposed in the discussion.

Comment 19:

The inclusion of analyses comparing ED diagnostic groups, irrespective of food addiction cluster, appeared to detract from the comparisons across clusters and made it unclear what additional value the clusters provide.

Response. We appreciate the Reviewer’s constructive comment. We agree that the comparisons between the clusters for the variables measures at baseline and the distribution of the diagnostic subtype is not included as a specific objective in this work. In fact, the original manuscript (Jiménez et al., 2019) published evidence regarding the differences between the clusters in these variables, since this information was necessary to adequately describe the clinical profiles associated with each of the empirical groups. However, since this new study did not analyze the complete data set used for the clustering, it seemed relevant to provide the results of the comparison between the empirical groups at the beginning of the treatment and in the ED type, to verify that the composition of the clusters used for this study was not altered and to facilitate a better definition of the profiles contained in each cluster. We have now described and justified the comparisons of the study in the statistical analysis section:

“Firstly, the empirical clusters compared in this study were compared for the measures assessed at baseline, and for the ED diagnostic subtype, to provide information about the specific profile associated to each cluster. Next, it was assessed the discriminative capacity of the empirical clusters on the treatment response”.

Comment 20:

Drop out seems qualitatively different from completing treatment. It is also more common that drop out and remission are examined in separate analyses. Therefore, the authors could bolster their rationale for including treatment drop-out in the same analyses as treatment remission.

Response. Thank you for this interesting report. We agree with the reviewer that there is no consensus of how therapeutic efficacy should be assessed in the study of eating disorders. In this study, we have used a definition typically employed in the research area, a classification which includes the “dropout” as a frequent response for the treatment as you can see in other studies, that are also cited in the manuscript text, in the last paragraph of treatment outcome assessment section (see for example, Agüera et al., 2017; Riesco et al., 2018).  

Comment 21:

According to Figure 2, it appears that a large percentage (close to 50%) of the sample dropped out prior to the end of treatment. Could the authors provide an overall sample breakdown of how many patients were included in each outcome group? This could help the reader better understand how treatment outcome in the present study might compare to prior studies.

Response: Thank you for the comment, that information it is in Table 2.

Discussion

Comment 22:

It appears that the 3 food addiction clusters did not replicate perfectly from the prior study. For example, food addiction was highest in the “moderate” cluster and did not significantly differ between the highest and lowest severity clusters. This calls to question the replicability and generalizability of these previously derived clusters to an eating disorder population. It is possible that an improved understanding of how the clusters were originally derived would mitigate this concern. Currently, discussion as to why the food addiction clusters may not have replicated seems warranted.

Response: Thank you very much for your comment, more information of the cluster analysis in the first study has already been added to the manuscript, as you can see as an answer to previous comments. About the replicability, actually the specific aspect that you mention, this is, FA severity, and how it is distribute in the 3 clusters, is equal in both studies.

Comment 23:

Given the pattern of findings, it is unclear whether the clusters are driving treatment outcome. Rather, it appears that food addiction severity maps onto treatment drop out and patients with lower symptom severity at the start of treatment achieve full remission. These findings seem to diminish the utility of the food addiction clusters. More intensive analyses testing the added clinical and predictive value of the clusters, above and beyond these factors, could increase confidence in the author’s conclusions.

Response. We appreciate this helpful suggestion. As we indicated in a previous response, we have clarified this point in the discussion section (see also our second response for the overall comments).

Comment 24:

It appears that one major limitation of the present study is the lack of a control group of individuals with eating disorders who do not screen positive for food addiction. The addition of this group could have provided more confidence that food addiction clustering, rather than eating disorder diagnosis or symptom severity, impacts treatment outcome.

Response: Thank you very much for the comment, however, the nature of the study was to understand the characteristics of population with FA positive, that it why both studies are complemented. However, we will take this into consideration for future studies.

Reviewer 2 Report

1) Why the age of BED group was almost 10 years higher than other groups?

Moreover, was there any possibility that this point will affect the analysis?

2) Please specify your research design (intervention study, observational study, etc) and check the guideline. For example, if this study was observational study, please add some parts following the STROBE guideline, such as sample size estimation, bias, validity, reliability, etc. Please indicate one by one in the text and/or adding STROBE guideline checklist (e.g. "The authors did not perform the sample size estimation."

3) Please specify whether the same equipment was used to measure weight and height, and include the name of the device.

Author Response

Reviewer #2:

Comment 1:

1) Why the age of BED group was almost 10 years higher than other groups? Moreover, was there any possibility that this point will affect the analysis?

Response: Thank you so much for your positive feedback about our work. Regarding this precise comment, yes, actually there are studies in the literature that confirm that the age of onset of the BED could be older than in other ED subtypes (PMID: 35044479). However, it is not possible that this fact affect the analysis, because the clustering process was originally performed for a large set of indicators measures in the baseline (previously to the intervention), which included precisely the ED severity level along with other variables (the comorbid psychopathological status, the personality traits and the diagnostic subtype, the age of onset, etc.). In this sense, belonging to a specific cluster provides wide information about the patients’ "profile" in a broad group of clinical indicators, that are already there, and do not influence the analysis, but they give us information needed, for example, define a better treatment approach,  as we aim with this both studies.

Comment 2:

2) Please specify your research design (intervention study, observational study, etc) and check the guideline. For example, if this study was observational study, please add some parts following the STROBE guideline, such as sample size estimation, bias, validity, reliability, etc.

Please indicate one by one in the text and/or adding STROBE guideline checklist (e.g. "The authors did not perform the sample size estimation."

Response. Thanks for this interesting remark. Our study was conducted following the guidelines of the APA-7 Style-Guide, published by the American Psychiatric Association and considered a standard guide in the psychopathology area. Certainly, a sample calculation was not provided, as the study included all participants who met the inclusion criteria and who had information for the outcome of the intervention. The comparison between relatively small sample groups is included as a limitation in the study

The study was only performed in women patients and the size of the sample was con-siderably reduced due to the exclusion of patients with obesity only (as the aim of this paper was on ED treatment response). These aspects affect the generalization of the results and future studies should consider them”.

Comment 3:

3) Please specify whether the same equipment was used to measure weight and height, and include the name of the device.

Response: Thank you very much to give us the opportunity to specify this information, we have added it to the first paragraph of the assessment section as follows:

Taking weight and height measures in the first visit to our center by trained staff, using the same device to all patients: Tanita MC 780-S MA portable scale: With segmental multifrequency.

Reviewer 3 Report

The manuscript is an interesting evaluation of the treatment outcome in patients with ED, looking at the role of FA. The paper used the clinical group of another paper, but the analyses are different, and it is clearly stated. I do not have particular concerns about the paper. However, I have some comments for the authors that should be taken into consideration for the benefit of the manuscript:

  • please briefly clarify how the clusters were evaluated in the methods. A reader could not be able to find the other paper. 
  • The study has high levels of dropout. Is there any hypothesis about this aspect?
  • Impulsivity should have a role in FA, and preliminary data have shown that CBT approaches could improve this aspect (see https://doi.org/10.1038/s41598-021-87231-w). However, this aspect is poorly evaluated in your analysis. 
  • Is there any suggestion for the ED field that could be driven by your study due to the discussed nature of FA? 

Author Response

Reviewer #3:

The manuscript is an interesting evaluation of the treatment outcome in patients with ED, looking at the role of FA. The paper used the clinical group of another paper, but the analyses are different, and it is clearly stated. I do not have particular concerns about the paper. However, I have some comments for the authors that should be taken into consideration for the benefit of the manuscript:

Comment 1:

Please briefly clarify how the clusters were evaluated in the methods. A reader could not be able to find the other paper. 

Response. We agree this matter requires further consideration. We have now included the clustering procedure used in the original study:

“The identification of the empirical clusters in this prior research was done through two-step-cluster procedure, using the log-likelihood distance measure (adequate for both quantitative and categorical indicators), and combining a multinomial probability mass function (non-metric data) and a normal density function (metric data). The clustering process was also based on an automatic selection of the number of clus-ter-classes, based on a large set of indicator variables including the ED severity level, global psychopathological state, personality profile and the diagnostic subtype”

Comment 2:

The study has high levels of dropout. Is there any hypothesis about this aspect?

Response: It is interesting that you point this out. Indeed, FA is highly related with other variables that contribute with low adherence to treatment, for example, impulsivity, less self-directedness, among others, that is why we attempt to identify subgroups, and to know their characteristics, and to provide to the patients the best treatment that could reduce the dropouts or bad compliance of treatment.

Comment 3:

Impulsivity should have a role in FA, and preliminary data have shown that CBT approaches could improve this aspect (see https://doi.org/10.1038/s41598-021-87231-w). However, this aspect is poorly evaluated in your analysis. 

Response: We agree with what you mention, impulsivity have an important role in FA, we do not consider this variable for the present analysis, but for sure will be included in future research. However, we have added the study you mention to us in the discussion of the manuscript.

As well, BED patients and FA patients present high levels of impulsivity, therapies aimed to reduce food related impulsivity could be implemented as well (Schag et al., 2021).

Comment 4:

Is there any suggestion for the ED field that could be driven by your study due to the discussed nature of FA? 

Response: Thank you very much for give us the opportunity to explain the implication of our study even a little bit more. The most important contribution of this work, is to offer a phenotypic characterization of the ED patients with FA, in order to suggest better treatment approaches according with the characteristics of each cluster, as we mention in the discussion and conclusion:

The Dysfunctional cluster may benefit from treatments that target aspects of high severity ED symptoms and psychological distress; the moderate cluster may be specifically benefited by a focused treatment for the reduction of FA symptoms; finally, the functional cluster could continue with traditional approaches.